# ON DIFFERENTIAL PRIVATE $\ell_1$, $\ell_2$ AND $\ell_p^p$ DISTANCE QUERIES

## ABSTRACT

We introduce a refined differentially private (DP) data structure for kernel density estimation (KDE) with $\ell_1$, $\ell_2$ and $\ell_p^p$ distance kernels. This new DP data structure offers not only improved privacy-utility tradeoff but also better query efficiency over prior results. Specifically, we study the mathematical problem: given a similarity function $f$ (or DP KDE) and a private dataset $X \subset \mathbb{R}^d$, our goal is to preprocess $X$ so that for any query $y \in \mathbb{R}^d$, we approximate $\sum_{x \in X} f(x, y)$ in a differentially private fashion. The best previous algorithm for $f(x, y) = \|x - y\|_1$ is the node-contaminated balanced binary tree by [Backurs, Lin, Mahabadi, Silwal, and Tarnawski, ICLR 2024]. Their algorithm requires $O(nd)$ space and time for preprocessing with $n = |X|$. For any query point, the query time is $\alpha^{-1} d \log^2 n$, with an multiplicative error guarantee of $(1 + \alpha)$-approximation and an additive error guarantee of $O(\epsilon^{-1} \alpha^{-0.5} d^{1.5} R \log^{1.5} n)$.

In this paper, we use the same space and pre-processing time, improve the best previous result [Backurs, Lin, Mahabadi, Silwal, and Tarnawski, ICLR 2024] in three aspects

- We reduce query time by $\alpha^{-1} \log n$ factor
- We improve the approximation ratio from $1 + \alpha$ to $1$
- We reduce the error dependence by a factor of $\alpha^{-0.5}$

From a technical perspective, our method of constructing the search tree differs from previous work [Backurs, Lin, Mahabadi, Silwal, and Tarnawski, ICLR 2024]. In prior work, for each query, the answer is split into $\alpha^{-1} \log n$ numbers, each derived from the summation of $\log n$ values in interval tree countings. In contrast, we construct the tree differently, splitting the answer into $\log n$ numbers, where each is a smart combination of two distance values, two counting values, and $y$ itself. We believe our tree structure may be of independent interest.

## 1 INTRODUCTION

We propose a refined differentially private (DP) data structure for DP kernel density estimation, offering improved privacy-utility tradeoff over prior results without compromising efficiency. Let $X \subset \mathbb{R}^d$ be a private dataset, and let $f(x, y) : \mathbb{R}^d \times \mathbb{R}^d \to \mathbb{R}$ be a similarity function[1], such as a kernel or distance function, between a user query $y \in \mathbb{R}^d$ and a private data point $x \in X$.

**Problem 1.1** (DP Kernel Density Estimation Query). *The DP Kernel Density Estimation (KDE) query problem aims for an algorithm outputting a private data structure $D_X : \mathbb{R}^d \to \mathbb{R}$, that approximates the map $y \mapsto \sum_{x \in X} f(x, y)$ in a DP fashion[2]. Especially, we require $D_X$ to remain private with respect to $X$, regardless of the number of queries.*

Problem 1.1 is well-studied (Hall et al., 2013; Huang & Roth, 2014; Wang et al., 2016; Aldà & Rubinstein, 2017; Alman et al., 2020; Coleman & Shrivastava, 2021; Aggarwal & Alman, 2022; Qin et al., 2022; Alman & Song, 2023; Gao et al., 2023; Wagner et al., 2023; Alman & Song, 2024; Hu et al., 2024; Li et al., 2024a; Backurs et al., 2024) due to its broad applicability and its importance in productizing large foundation models (Lin et al., 2024; Xie et al., 2024a). What makes this problem interesting is its generic abstraction of many practical DP challenges in modern machine learning.

---

[1]In this work, we use "kernel query" and "distance query" interchangeably.

[2]For a given dataset $X$ and a given query $y$, an KDE query computes an approximation to $\sum_{x \in X} f(x, y)$.

These include generating synthetic data similar to a private dataset (Lin et al., 2020; Li et al., 2022; Yu et al., 2022; Yin et al., 2022), and selecting a similar public dataset for pre-training ML models (Hou et al., 2023; Yu et al., 2024a; Yue et al., 2023). Such problem becomes more prevalent in this era of large foundation models (Lin et al., 2024; Xie et al., 2024a). Essentially, these problems involve computing the similarity between a private dataset (i.e., $X$) and a processed data point (i.e., query $y$), thus falling under Problem 1.1.

In this work, we focus on the $\ell_1$ kernel[3] (i.e., $f(x, y) = \|x - y\|_1$). By far, the algorithm with the best privacy-utility tradeoff and query efficiency for Problem 1.1 is by Backurs, Lin, Mahabadi, Silwal, and Tarnawski (Backurs et al., 2024). They propose a DP data structure via a node-contaminated balanced binary tree for $\ell_1$ kernel. To be concrete, we begin by defining the similarity error between two data structures and then present their results.

**Definition 1.2** (Similarity Error between Two Data Structures). *For a fixed query, let $A$ represent the value output by our private data structure, and let $A'$ represent the true value. We say that $A$ has an error of $(M, Z)$ for $M \geq 1$ and $Z \geq 0$, if $\mathbb{E}[|A - A'|] \leq (M - 1)A' + Z$. This implies a relative error of $M - 1$ and an additive error of $Z$. The expectation is taken over the randomness used by the data structure.*

Let $n := |X|$ be the size of the dataset, $X_{i,j}$ be the $j$-th dimension of the $i$-th data point, $R := \max_{i,j}(X_{i,j})$ be the max value of each data entries from each dimension, and $\alpha \in [0, 1]$ be a parameter of the data structure selected before calculation. For $\ell_1$ kernel, Backurs et al. (2024) give the error bound $(1 + \alpha, \alpha^{-0.5}\epsilon^{-1}Rd^{1.5}\log^{1.5} n)$ through their DP data structure.

Compared to the results of (Backurs et al., 2024), we provide a refined DP data structure and improves both privacy-utility tradeoff and query efficiency. Specifically, we not only improve the $\ell_1$ error bound to $(1, \epsilon^{-1}Rd^{1.5}\log^{1.5} n)$, but also the query time from $O(\alpha^{-1}d\log^2 n)$ to $O(d\log n)$. This makes our new DP data structure the latest best algorithm for Problem 1.1:

**Theorem 1.3** (Informal Version of Theorem 3.11). *Given a dataset $X \subset \mathbb{R}^d$ with $|X| = n$. There is an algorithm that uses $O(nd)$ space to build a data-structure which supports the following operations:*

- **INIT**$(X, \epsilon)$*. It takes dataset $X$ and privacy parameter $\epsilon$ as input and spends $O(nd)$ time to build a data-structure.*

- **QUERY**$(y \in \mathbb{R}^d)$*. It takes $y$ as input and spends $O(d\log n)$ time to output a scalar $A$ such that*
$$\mathbb{E}[|A - A'|] \leq O(\epsilon^{-1}Rd^{1.5}\log^{1.5} n).$$

Notably, our improvements stem from a key observation that — in the balanced binary tree data structure proposed by Backurs et al. (2024), each query answer is split into $\alpha^{-1}\log n$ values, derived from the summation of $\log n$ values in interval tree counts. We deem that this splitting is not essential and can be achieved through simple and efficient preprocessing steps.

To this end, we introduce a novel data representation in the tree nodes, where each node stores the sum of distances from one point to multiple points. This allows us to split the answer into only $\log n$ values, each being a smart combination of two distance values, two count values, and $y$ itself. This additional information enhances each query calculation, leading to an improved privacy-utility tradeoff (Theorem 3.11 and Lemma 3.5) and faster query time (Lemma 3.3).

Lastly, we remark that our $\ell_1$ results are transferable to other kernels via the provably dimensional reduction recipe of (Backurs et al., 2024). To showcase this, we generalize our results to $\ell_2$ and $\ell_p^p$ kernels following (Backurs et al., 2024). Notably, these results also improve upon the best previous results reported in (Backurs et al., 2024) for $\ell_2$ and $\ell_p^p$ with $p \in [1, n]$. For details, see Appendix E and F, particularly the comparison Tables 2 and 3.

**Related Work.** We defer related work discussion to Appendix B due to page limits.

**Organization.** Section 2 includes the preliminaries, summarizes the prior best DP data structure by (Backurs et al., 2024), and provides a high-level overview of our main results. Section 3 presents

---

[3]We remark that $\ell_1$ query distance is fundamental. Namely, it is easy to generalize the $\ell_1$ results to a wide range of kernels and distance functions following the provably dimensional reduction recipe of (Backurs et al., 2024). See Appendix E and F for improved DP KDE results with $\ell_2$ and $\ell_p^p$ kernels.

our new data structure for DP $\ell_1$ kernel estimation along with its theoretical analysis. Appendices E and F extends the $\ell_1$ results to $\ell_2$ and $\ell_p^p$ kernels. Appendix A includes proof-of-concept experiments. Lastly, Section 4 concludes the paper.

| References | Approx. Ratio | Error | Query Time | Init time | Space |
|---|---|---|---|---|---|
| (Backurs et al., 2024) | $1 + \alpha$ | $\alpha^{-0.5}\epsilon^{-1}Rd^{1.5}\log^{1.5} n$ | $\alpha^{-1}d\log^2 n$ | $nd$ | $nd$ |
| Theorem 1.3 | 1 | $\epsilon^{-1}Rd^{1.5}\log^{1.5} n$ | $d\log n$ | $nd$ | $nd$ |

Table 1: Comparison of $\|x - y\|_1$ Results with Best Known Algorithm by Backurs et al. (2024). We ignore the big-O notation in the table, for simplicity of presentation. We use $n$ to denote the number of points in dataset. We use $d$ to denote the dimension for each point in the dataset. We assume all the points are bounded by $R$. We use the $\epsilon$ to denote $\epsilon$-DP.

## 2 Preliminaries

**Notation.** For a positive integer $n$, we use $[n]$ to denote $\{1, 2, \cdots, n\}$. For a vector $x$, we use $\|x\|_1 := \sum_{i=1}^{n} |x_i|$ to denote its $\ell_1$-norm. We use $\mathbb{E}[\cdot]$ to denote the expectation. We use $\mathrm{Var}[\cdot]$ to denote the variance. We use $\Pr[\cdot]$ to denote the probability. We use $\mathsf{Laplace}(\lambda)$ random variable with parameter $\lambda$. It is known that $\mathbb{E}[\mathsf{Laplace}(\lambda)] = 0$ and $\mathrm{Var}[\mathsf{Laplace}(\lambda)] = 2\lambda^2$.

### 2.1 Differential Privacy

For completeness, we state the definitions of differential privacy (Dwork et al., 2006; 2010b).

**Definition 2.1** (Pure/Approximate Differential Privacy). *A randomized algorithm $M : \mathcal{X}^n \to \mathcal{Y}$ satisfies ($\epsilon$, $\delta$)-differential privacy if, for all $x, x' \in \mathcal{X}^n$ differing on a single element and for all events $E \subset \mathcal{Y}$, we have $\Pr[M(x) \in E] \leq e^\epsilon \cdot \Pr[M(x') \in E] + \delta$*

For special case of ($\epsilon$, 0)-differential privacy, we use pure or pointwise $\epsilon$-differential privacy to denote it. For the other case of $\delta > 0$, we denote it as approximate differential privacy.

Next, we introduce a common tool in DP proofs. For multiple independent DP functions, the next lemma provides an estimation on the privacy of their composition:

**Lemma 2.2** (Advanced Composition Starting from Pure DP (Dwork et al., 2010b)). *Let $M_1, ..., M_k : X^n \to Y$ be randomized algorithms, each of which is ($\epsilon, \delta$)-DP. Define $M : X^n \to Y^k$ by $M(x) = (M_1(x), ..., M_k(x))$ where each algorithm is run independently. Then $M$ is ($\epsilon', \delta$)-DP for any $\epsilon, \delta > 0$ and*

$$\epsilon' = k\epsilon^2/2 + \epsilon\sqrt{2k\log(1/\delta)}.$$

*For $\delta = 0$, $M$ is $k\epsilon$-DP.*

### 2.2 (Backurs et al., 2024)'s DP Data Structure: Node-Contaminated Balanced Tree

Backurs et al. (2024) propose a data structure for general high-dimensional (e.g., $d$-dimensional) $\ell_1$ kernel via 1-dimensional decomposition:

$$\sum_{x \in X} \|x - y\|_1 = \sum_{i=1}^{d} \sum_{x \in X} |x_i - y_i|.$$

Namely, they create a DP data structure for a $d$-dimensional dataset $X$ by considering $d$ copies 1-dimensional DP data structures (i.e., $\sum_{x \in X} |x_i - y_i|$ for $i \in [d]$). For each of these 1-dimensional DP data structures, they employ a node-contaminated balanced tree algorithm as follows.

Let $n = |X|$ be the size of the dataset $X$. Considering all input values are integer multiples of $R/n$ in $[0, R]$,[4]

1. They use a classic balance binary tree of $L$ layers in one dimension: a binary tree where each non-leaf node has exactly two children, and all leaf nodes are at the same depth, $L$. The tree has $2^L - 1$ total nodes, with $2^{L-1}$ leaf nodes and $L$ levels from the root (level 1) to the leaves (level $L$). Each level $k$ contains $2^{k-1}$ nodes.

---

[4]For general continuous data, they achieve this by rounding all data points to some integer multiples of $R/n$.

2. They assign the leaves to correspond to these multiples (of $R/n$ in $[0, R]$) and store the number of dataset points at each position, while internal nodes store the sum of their children.

3. It's well-known that any interval query can be answered by summing values from $O(\log n)$ tree nodes. This provide an efficient (*noise-less*) query operation.

4. To ensure DP, they propose a noisy version of the tree by injecting noise onto each node, i.e., a *node-contaminated* balanced tree. By above, changing any data point affects only $O(\log n)$ counts in the tree, each by at most one (from leaf to root), which bounds the sensitivity of the data structure.

To highlight the significance of this work — removing the $\alpha$ dependence in the result and eliminating a $\log n$ factor in the query (see Table 1) — we make the following remarks on (Backurs et al., 2024).

- **Query Time.** For each query $y$, in order to report the answer, they need to take summation of $O(\alpha^{-1} \log n)$ numbers where each number from each region $(y+(1+\alpha)^i, y+(1+\alpha)^{i+1}]$ (see line 7 in Algorithm 3 in (Backurs et al., 2024)). In particular, for each region, they need to run a interval query which consists of $\log n$ numbers (each number is from a tree node, see Algorithm 2 in (Backurs et al., 2024)). That's why the their algorithm has $\alpha^{-1} \log^2 n$ dependence.

- **Error Bound and Accuracy.** Due to the region $(y + (1 + \alpha)^i, y + (1 + \alpha)^{i+1}]$ can only approximate the number within $(1 + \alpha)$ approximation. That is why the final error is proportional to $\alpha^{-1}$ and accuracy is losing a $(1 + \alpha)$ relative error. Since the values are in range $[0, R]$, thus the error bound will be linearly depend on $R$ according to how they construct the tree and alayze the error.

- **Sensitivity and DP Guarantee.** Since the tree is analyzing the counting problem, thus the sensitivity is $O(\log n)$ because change one point, will affect $O(\log n)$-levels stored counts.

### 2.3 HIGH-LEVEL OVERVIEW OF OUR DP DATA STRUCTURE

We improve (Backurs et al., 2024) by providing a refined balanced tree for DP 1-dimensional $\ell_1$ distance query. Specifically, we introduce a new data representation on each node of the balanced binary tree as follows.

Let $n = |X|$ be the size of the dataset $X$ and $\{x_k\} \in [0, R)$ for $k \in [n]$. (Note that unlike (Backurs et al., 2024), we don't assume $x_k$ is a multiple of $R/n$, because our improved algorithm is compatible with floating-point values of $x_k$.)

1. We use a balanced binary tree where in each node stores both distance $s$ and the counts $c$. Here, $s$ denotes the pre-summation of certain distances, and $c$ denotes the pre-counting.

2. We make a key observation (Lemma 3.1) that these values ($s$ and $c$) are not depending on query $y$, while they are critical components of the final answer (the $\ell_1$ kernel $\sum_{x \in X} \|x - y\|_1$). This motivate us to construct the answer by using $L = \log n$ layers result, where each layer result is a smart combination of $y, s_1, s_2, c_1, c_2$ (i.e., $(s_1 - s_2 + y \cdot c_1 - y \cdot c_2)$ ).

3. To ensure DP, we need a noise version of the tree, due to $s$ and $c$ have different sensitivities, thus we need to add different levels of noise for $s$ and $c$.

A few remarks are in order:

- **Query Time.** For each query $y$, to report the answer, we first extract four $L = O(\log n)$ numbers along with $y$ and apply a special function: $(s_1 - s_2 + y \cdot (c_1 - c_2))$ (see our key observation in Lemma 3.1).

- **Error Bound and Accuracy.** Since our algorithm does not use the $(1+\alpha)^i$ region concept, we avoid the $\alpha^{-1}$ error. As a result, our error is purely additive, with no relative error.

- **Sensitivity and DP Guarantee.** Since $s$ approximates the summation of distances, the sensitivity is $O(LR)$, where each node in one of the $L$ layers contributes a factor of $R$. On the other hand, since $c$ counts the points, its sensitivity is $O(L)$, with each node in the $L$ layers contributing an $O(1)$ factor. Previous work by Backurs et al. (2024) only use $y$ to determine which region to count points, but our algorithm uses $y$ to construct the final answer. Since $y \leq R$, this explains the $R$ gap between the noise levels in $s$ and $c$.

## 3 A REFINED DIFFERENTIALLY PRIVATE DATA STRUCTURE

We propose a new data structure with $\epsilon$-differential privacy that achieves $O(\epsilon^{-1} R \log^{1.5} n)$ additive error. Our method improves the balanced binary tree structure proposed by Backur, Lin, Mahabadi, Silawal, and Tarnawsk (Backurs et al., 2024) in both DP guarantee and efficiency. Our data structure adapts a new data representation on each node that simplifies each query without losing privacy guarantees.

### 3.1 KEY OBSERVATION AND NEW DATA STRUCTURE FOR ONE DIMENSIONAL $\ell_1$ DISTANCE QUERY

For simplicity, we consider the 1-dimensional distance query problem with $\ell_1$ kernel distance. Note that any high-dimensional distance query problem can be reduced to this case via the 1-dimensional decomposition discussed in Section 2.2 or in (Backurs et al., 2024).

When doing queries, we observe that

**Lemma 3.1.** *For a collection of values $\{x_1, x_2, \cdots, x_n\} \subset \mathbb{R}$ and a value $y$, we define two sets*

$$S_+ := \{k \in [n] \ : \ x_k > y\}$$
$$S_- := \{k \in [n] \ : \ x_k < y\}.$$

*It holds*

$$\sum_{k=1}^{n} |x_k - y| = \underbrace{(\sum_{k \in S_+} x_k)}_{s_{\text{right}}} - \underbrace{(\sum_{k \in S_-} x_k)}_{s_{\text{left}}} + y \cdot \underbrace{|S_-|}_{c_{\text{left}}} - y \cdot \underbrace{|S_+|}_{c_{\text{right}}}.$$

*Proof.*

$$\sum_{k=1}^{n} |x_k - y|$$
$$= \sum_{k \in S_+} (x_k - y) + \sum_{k \in S_-} (y - x_k)$$
$$= (\sum_{k \in S_+} x_k) - (\sum_{k \in S_-} x_k) + y \cdot (\sum_{k \in S_-} 1) - y \cdot (\sum_{k \in S_+} 1)$$
$$= (\sum_{k \in S_+} x_k) - (\sum_{k \in S_-} x_k) + y \cdot |S_-| - y \cdot |S_+|.$$

This completes the proof. $\qquad\square$

Lemma 3.1 provides a neat decomposition of the $\ell_1$ distance query into four components. This motivates us to design a new DP data structure that pre-computes these terms and storing them in a balanced tree for possible algorithmic speedup. Surprisingly, this preprocessing not only accelerates query time but also improves the privacy-utility tradeoff. The following discussion illustrates this.

To calculate each part efficiently, we define our data representation on a balanced binary tree as follows:

- **Dataset**: Given a dataset $X := \{x_k\}_{k=1}^{n}$ containing $n$ values in the range $[0, R)$, we build a balanced binary tree using $X$.
- **Tree Structure**:
    - Let $L$ denote the total number of layers in the tree.
    - At the $l$-th layer, there are exactly $2^l$ nodes.
- **Node Representation**:
    - Each node represents a consecutive interval of $X$.
    - The interval for the $j$-th node at the $l$-th layer is defined as $I_{l,j} := [(j - 1) \cdot \frac{R}{2^{l-1}}, j \cdot \frac{R}{2^{l-1}})$.
    - The left and right children of $I_{l,j}$ are $I_{l+1,2j-1}$ and $I_{l+1,2j}$, respectively.

Each node $I_{l,j}$ stores two values: $c_{l,j}$ and $s_{l,j}$. Here:

- $c_{l,j} := |\{x_k : x_k \in I_{l,j}\}|$ represents the count of data points in the interval.

- $s_{l,j} := \sum_{x_k \in I_{l,j}} x_k$ represents the sum of the data points' values in the interval.

After calculating each $c_{l,j}$ and $s_{l,j}$, we add independent noise drawn from $\mathsf{Laplace}(L/\epsilon)$ to each $c$ value and from $\mathsf{Laplace}(LR/\epsilon)$ to each $s$ value. The INIT algorithm (Algorithm 1) outlines the construction of our data structure.

In the QUERY algorithm (Algorithm 2), we sum the $s$ and $c$ values at the relevant nodes. Here:

- $s_{\text{left}}$ represents $\sum_{k \in S_-} x_k$
- $c_{\text{left}}$ represents $|S_-|$
- $s_{\text{right}}$ represents $\sum_{k \in S_+} x_k$
- $c_{\text{right}}$ represents $|S_+|$

For each query, we first locate the leaf that $y$ belongs to, then traverse the tree from the bottom up, summing the values of the left and right nodes.

Next, we prove the time complexity, differential privacy property and error of results of out algorithm respectively.

---

**Algorithm 1** Building Binary Tree

1: **data structure** FASTERTREE ▷ Theorem 3.8
2: **members**
3:    $T := \{c, s\}$ ▷ $c_{i,j}$ and $s_{i,j}$ represents the count and sum of node defined above
4: **end members**
5: **procedure** INIT$(X, n, \epsilon, L)$ ▷ Lemma 3.2, Lemma 3.5
6:    **for** each $x_k$ in $X$ **do**
7:       find the leaf node $j$ of layer $L$ that $x_k$ is in the interval $I_{L,j} = [(j-1) \cdot \frac{R}{2^{L-1}}, j \cdot \frac{R}{2^{L-1}})$
8:       $c_{L,j} \leftarrow c_{L,j} + 1$
9:       $s_{L,j} \leftarrow s_{L,j} + x_k$
10:    **end for**
11:    **for** each layer $l$ from $L-1$ to $1$ **do**
12:       **for** each node $j$ in layer $l$ **do**
13:          $c_{l,j} \leftarrow c_{l+1,2j-1} + c_{l+1,2j}$
14:          $s_{l,j} \leftarrow s_{l+1,2j-1} + s_{l+1,2j}$
15:       **end for**
16:    **end for**
17:    Add independent noises drawn from $\mathsf{Laplace}(LR/\epsilon)$ to each $s_{l,j}$
18:    Add independent noises drawn from $\mathsf{Laplace}(L/\epsilon)$ to each $c_{l,j}$
19:    $T \leftarrow \{c, s\}$
20: **end procedure**
21: **end data structure**

---

### 3.2 TIME COMPLEXITY

Here we compute the init and query time of Algorithm 1 and 2, respectively.

**Lemma 3.2** (Init Time). *If the total layer $L = \log(n)$, the running time of* INIT *(Algorithm 1) is* $O(n)$.

*Proof.* By definition, on $l$-th layer, there are $2^l$ nodes. Therefore, the total number of nodes is $\sum_{l=1}^{L} 2^l = 2^{L+1} - 1$. When $L = \log(n)$, the total number of nodes on the tree is $O(n)$. In INIT (Algorithm 1), we iterate over $n$ data points, then iterate all nodes. Therefore the total time complexity of pre-processing is $O(n)$. □

**Lemma 3.3** (Query Time). *If the total layer $L = \log(n)$, the time* QUERY *function (Algorithm 2)* $O(\log(n))$ *time.*

*Proof.* For each single query, we iterate the node that $y$ belongs to on each layer. The total layer number is $\log(n)$, so the time complexity of each query is $O(\log(n))$. □

**Remark 3.4** (Comparing with Prior Work). *For a dataset $X \subset \mathbb{R}^d$ of size $n = |X|$ and a parameter $\alpha \in [0,1]$ selected before calculation, Lemma 3.3 improves the prior best query time in (Backurs et al., 2024) by a factor of $\alpha^{-1} \log n$.*

---

**Algorithm 2** One Dimensional Distance Query

---

1: **data structure** FASTERTREE                                           ▷ Theorem 3.8
2: **procedure** QUERY($y \in \mathbb{R}$)                              ▷ Lemma 3.3, Lemma 3.6
3:      $s_{\text{left}}, c_{\text{left}}, s_{\text{right}}, c_{\text{right}} \leftarrow 0$                            ▷ As previously defined
4:      **for** each layer $l$ from 2 to $L$ **do**
5:          find the node $j = \lceil \frac{2^{l-1}}{R} y \rceil$ in layer $l$ so that $y$ is in the interval $I_{l,j} = [(j-1) \cdot \frac{R}{2^{l-1}}, j \cdot \frac{R}{2^{l-1}})$
6:          **if** $j \mod 2 = 0$ **then**                ▷ if node $j$ is the right child of its parent
7:              $c_{\text{left}} \leftarrow c_{\text{left}} + c_{l,j-1}$
8:              $s_{\text{left}} \leftarrow s_{\text{left}} + s_{l,j-1}$
9:          **else**                                  ▷ if node $j$ is the left child of its parent
10:             $c_{\text{right}} \leftarrow c_{\text{right}} + c_{l,j+1}$
11:             $s_{\text{right}} \leftarrow s_{\text{right}} + s_{l,j+1}$
12:          **end if**
13:      **end for**
14:      **return** $s_{\text{left}} - s_{\text{right}} + y \cdot c_{\text{right}} - y \cdot c_{\text{left}}$             ▷ Lemma 3.1
15: **end procedure**
16: **end data structure**

---

### 3.3 PRIVACY GUARANTEES

Here we provide our DP guarantee of our proposed DP data structure (i.e., Algorithm 1).

**Lemma 3.5** (Differential Privacy). *The data structure* FASTERTREE *returned by* FASTERTREE.INIT *(Algorithm 1) is $\epsilon$-DP.*

*Proof.* For binary tree of $L$ layers, there are $2^{L+1} - 1$ nodes. On each node, there are 2 values $s$ and $c$. We consider $c$ and $s$ separately.

Let the functions $F_c(X)$ and $F_s(X) : [0, R)^n \to \mathbb{R}^{2^{L+1}-1}$ represent the mappings from dataset $X$ to $c$ and $s$, respectively. Next, we prove that both $F_c(X)$ and $F_s(X)$ are $\epsilon/2$-DP.

When changing each data point, only the nodes containing the point change their values. On each layer, there is at most one such node. Therefore, only $O(L)$ values change.

**Sensitivity for Storing Counts.** For $F_c$, each $c$ value changes by at most 1, so the sensitivity is $L$. Adding coordinate-wise Laplace noise with magnitude $\eta = O(2L/\epsilon)$ suffices to ensure $\epsilon/2$-DP using the standard Laplace mechanism.

**Sensitivity for Storing Distances.** For $F_s$, changing one data entry affects at most $O(L)$ values, and each value can be affected by at most $R$. Therefore, the sensitivity is $RL$. Adding coordinate-wise Laplace noise with magnitude $\eta = O(2RL/\epsilon)$ ensures $\epsilon/2$-DP.

By Lemma 2.2, the differential privacy parameter $\epsilon$ of the tree $T := (F_c(X), F_s(X))$ equals $2 \cdot \epsilon/2 = \epsilon$. This completes the proof.      $\square$

### 3.4 ERROR GUARANTEE

Here we provide the (data-structure) similarity error of our proposed DP data structure.

**Lemma 3.6** (Error Guarantee). *Let $X = \{x_i\}_{i \in [n]}$ be a dataset of $n$ one dimensional values $x_i \in [0, R)$. Let $A' = \sum_{i=1}^{n} |x_i - y|$ be the true distance query value. Let $A$ be the output of* QUERY *(Algorithm 2) with $L = \log(n)$. Then we have $\mathbb{E}[|A - A'|] \leq O(\log^{1.5}(n)R/\epsilon)$.*

*Proof.* We analyze the additive error of our algorithm. We notice that the difference between true distance and our output can be divided into two parts.

**Part I.** The first part is the data in the leaf node which contains query $y$. In QUERY(Algorithm 2), we ignore these data points. Let the leaf node $j$ of layer $L$ be the node that query point $y$ belongs to. The error is

$$\sum_{x_k \in [(j-1) \cdot R/2^L, j \cdot R/2^L)} |x_k - y|.$$

Since the distance between each data point and $y$ is no more than the length of the interval $R/2^L$, and their total number is no more than $n$. When $L = \log(n)$, this error is $O(R)$.

**Part II.** The second part is the Laplace noises. In our query, we add up $c$ and $s$ from $O(L)$ intervals. In these intervals, each contains a Laplace noise. Therefore, the total error is the sum of these noises:

$$\mathbb{E}[|A - A'|] \leq \mathbb{E}[|\sum_{i=1}^{L} \mathsf{Laplace}(RL/\epsilon) + y \cdot \sum_{i=1}^{L} \mathsf{Laplace}(L/\epsilon)|]$$

$$\leq \mathbb{E}[|\sum_{i=1}^{L} \mathsf{Laplace}(RL/\epsilon)|] + |y| \cdot \mathbb{E}[|\sum_{i=1}^{L} \mathsf{Laplace}(L/\epsilon)|],$$

where the second step follows from triangle inequality.

For a random variable $Q$ with $\mathbb{E}[Q] = 0$, using Fact D.1 we have $\mathbb{E}[|Q|] \leq \sqrt{\mathrm{Var}[Q]}$, thus

$$\mathbb{E}[|A - A'|] \leq (\mathrm{Var}[\sum_{i=1}^{L} \mathsf{Laplace}(RL/\epsilon)])^{1/2} + |y| \cdot (\mathrm{Var}[\sum_{i=1}^{L} \mathsf{Laplace}(L/\epsilon)])^{1/2}$$

$$\leq \sqrt{L \cdot \mathrm{Var}[\mathsf{Laplace}(RL/\epsilon)]} + |y| \cdot \sqrt{L \cdot \mathrm{Var}[\mathsf{Laplace}(L/\epsilon)]}$$

$$= \sqrt{L \cdot 2R^2 L^2/\epsilon^2} + |y| \cdot \sqrt{L \cdot 2L^2/\epsilon^2}$$

$$= \sqrt{2} \cdot (R + |y|)L^{1.5}/\epsilon$$

$$= O(\epsilon^{-1} R \log^{1.5}(n)).$$

where the third step follows from $\mathrm{Var}[\mathsf{Laplace}(x)] = 2x^2$.

The last step is because $y \in [0, R)$ and $L = \log(n)$. $\qquad\square$

**Remark 3.7** (Comparing with Prior Work). *For a dataset $X \subset \mathbb{R}^d$ of size $n = |X|$ and a parameter $\alpha \in [0, 1]$ selected prior to calculation, Lemma 3.6 improves the approximation ratio from $\alpha$ to 1 and reduces the error dependence by a factor of $\alpha^{-0.5}$ compared to (Backurs et al., 2024).*

### 3.5 ONE DIMENSIONAL DIFFERENTIALLY PRIVATE DATA STRUCTURE

Collectively, we present the init time, space, query time, approximation error, and privacy guarantees for the 1-dimensional DP data structure (Algorithm 1, 2) in the next theorem. As a reminder, we address the 1-dimensional distance query problem (Problem 1.1 with $d = 1$) using the $\ell_1$ kernel distance.

**Theorem 3.8** (1-Dimensional $\ell_1$ DP Distance Query). *Given a dataset $X \subset \mathbb{R}$ with $|X| = n$, there is an algorithm that uses $O(n)$ space to build a data structure (Algorithm 1, 2) which supports the following operations*

- **INIT**$(X, \epsilon)$. *It takes dataset $X$ and privacy parameter $\epsilon$ as input and spends $O(n)$ time to build a data-structure.*

- **QUERY**$(y \in \mathbb{R})$. *It takes $y$ as input and spends $O(\log n)$ time to output a scalar $A$ such that*
$$\mathbb{E}[|A - A'|] \leq O(\epsilon^{-1} R \log^{1.5} n).$$

*Furthermore, the data structure is $\epsilon$-DP.*

*Proof.* By Lemma 3.2, we prove the storage space, and the initialization time. By Lemma 3.3, we prove the query time of the data-structure. By Lemma 3.5, we prove the differential privacy guarantee. By Lemma 3.6, we prove the error guarantee after adding the noise. $\qquad\square$

### 3.6 HIGH DIMENSIONAL $\ell_1$ DISTANCE QUERY

We now generalize our 1-dimensional DP data structure to $d$-dimensional DP distance queries. Specifically, we apply our 1-dimensional DP data structure independently to each dimension to compute high-dimensional queries.

**Lemma 3.9** (Differential Privacy). *The data structure HIGHDIMFASTERTREE returned by HIGH-DIMFASTERTREE.INIT (Algorithm 3) is $\epsilon$-DP.*

*Proof.* The proof directly follows from calling Lemma 3.5 for $d$ times with replacing $\epsilon$ by $\epsilon/d$. Then using the DP composition Lemma. $\qquad\square$

---

**Algorithm 3** High Dimensional Distance Query

---

1: **Input:** $n$ $d$-dimensional points in the box $[0, R)^d$, privacy parameter $\epsilon$, query $y$.
2: **data structure** HIGHDIMFASTERTREE
3: **members**
4:     $T_i$ for $i \in [d]$                    ▷ $T_i$ represents a FasterTree in Algorithm 1 for $i-$th dimension.
5: **end memebers**
6: **procedure** INIT($X$)
7:     Initialize $d$ independent data structure $T_i$ for $i \in [d]$ with Algorithm 1. $T_i$ builds on $i$-th coordinate projections of the data points. Each data structure is $\epsilon/d$-private.
8: **end procedure**
9: **procedure** QUERY($y$)
10:     **Return** the sum of the outputs of Algorithm 2 from $T_i$ with query $y_i$ for $i \in [d]$.
11: **end procedure**
12: **end data structure**

---

**Lemma 3.10** (Error Guarantee). *Let $X = \{x_i\}_{i \in [n]}$ be a dataset of $n$ $d$-dimensional vectors $x_i \in [0, R)^d$. Let $A' = \sum_{i=1}^{n} |x_i - y|_1$ be the true distance query value. Let $A$ be the output of* QUERY *(Algorithm 3). Then we have $\mathbb{E}[|A - A'|] \leq O(\epsilon^{-1} R d^{1.5} \log^{1.5} n)$.*

*Proof.* The output of Algorithm 3 is the sum of each dimension. Let $A'_i$ be the output of $D_i$, $A_i$ be the true distance of the $i$-th dimension. The total error of Algorithm 3 is $|\sum_{i=1}^{d} (A_i - A'_i)|$. From the proof of Algorithm 2, we know $A_i - A'_i$ is the sum of $O(L)$ Laplace noises, where $L = O(\log n)$. Therefore, we have $\mathbb{E}[A_i - A'_i] = 0$, $\mathrm{Var}[A_i - A'_i] = O(\epsilon^{-2} d^2 R^2 \log^3(n))$. Using the independence property of $A_i - A'_i$ for all $i$, we bound $|\sum_{i=1}^{d} (A_i - A'_i)|$ by

$$\mathbb{E}[|\sum_{i=1}^{d} (A_i - A'_i)|] \leq (\mathrm{Var}[\sum_{i=1}^{d} (A_i - A'_i)])^{1/2}$$
$$= (d \, \mathrm{Var}[A_i - A'_i])^{1/2}$$
$$= O(d \cdot \epsilon^{-2} d^2 R^2 \log^3 n)^{1/2}$$
$$= O(\epsilon^{-1} d^{1.5} R \log^{1.5}(n)).$$

where the first step follows from Fact D.1. This completes the proof. $\square$

**Theorem 3.11** (*d*-Dimensional $\ell_1$ DP Distance Query, Formal version of Theorem 1.3). *Given a dataset $X \subset \mathbb{R}^d$ with $|X| = n$, there is an algorithm that uses $O(nd)$ space to build a data-structure (Algorithm 1, 2) which supports the following operations*

- **INIT**($X \subset \mathbb{R}^d, \epsilon \in (0, 1)$). *It takes dataset $X$ and privacy parameter $\epsilon \in (0, 1)$ as input and spends $O(nd)$ time to build a data structure.*

- **QUERY**($y \in \mathbb{R}^d$). *It takes $y$ as input and spends $O(d \log n)$ time to output a scalar $A \in \mathbb{R}$ such that $\mathbb{E}[|A - A'|] \leq O(\epsilon^{-1} R d^{1.5} \log^{1.5} n)$.*

*Proof.* By Lemma 3.9, we prove the privacy guarantees. By Lemma 3.10, we prove the error guarantee of the QUERY. This completes the proof. $\square$

## 4 CONCLUDING REMARKS

We present a refined data structure for differentially private kernel density estimation over the $\ell_1$ kernel. Our new DP data structure significantly improves the state of the art in both query efficiency and approximation quality. Specifically, it enhances the best-known node-contaminated balanced binary tree method (Backurs et al., 2024) in three aspects: faster queries (Lemmas 3.2 and 3.3), exact ratios (rather than approximate), and reduced error dependence (Lemma 3.6). Moreover, our approach achieves a strictly better privacy-utility tradeoff without incurring additional preprocessing overhead (Theorem 3.11). Empirically, our theory aligns with numerical experiments, especially Figures 1 and 2, which highlight the improved privacy-utility tradeoff and efficiency of our DP data structure, benchmarked against (Backurs et al., 2024). We also extend our results to $\ell_2$, and $\ell_p^p$ kernels in Appendices E and F. We defer numerical validations to Appendix A due to page limits.

## ETHIC STATEMENT

This paper does not involve human subjects, personally identifiable data, or sensitive applications. We do not foresee direct ethical risks. We follow the ICLR Code of Ethics and affirm that all aspects of this research comply with the principles of fairness, transparency, and integrity.

## REPRODUCIBILITY STATEMENT

We ensure reproducibility on both theoretical and empirical fronts. For theory, we include all formal assumptions, definitions, and complete proofs in the appendix. For experiments, we describe model architectures, datasets, preprocessing steps, hyperparameters, and training details in the main text and appendix. Code and scripts are provided in the supplementary materials to replicate the empirical results.

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

## IMPACT STATEMENT

Our results offer a combination of fast queries, tight accuracy, and robust privacy guarantees. These results are essential for large-scale data analysis scenarios where computational efficiency and data confidentiality are critical. By the formal nature of this work, we do not expect any immediate negative social impact.

## LLM USAGE DISCLOSURE

We used large language models (LLMs) to aid and polish writing, such as improving clarity, grammar, and conciseness. We also used LLMs for retrieval and discovery, for example exhausting literature to identify potential missing related work. All technical content, proofs, experiments, and results are original contributions by the authors.

## A  PROOF-OF-CONCEPT EXPERIMENTS

We validate the efficacy of our theory in both synthetic and real-world experiments.

### A.1  SYNTHETIC DATA

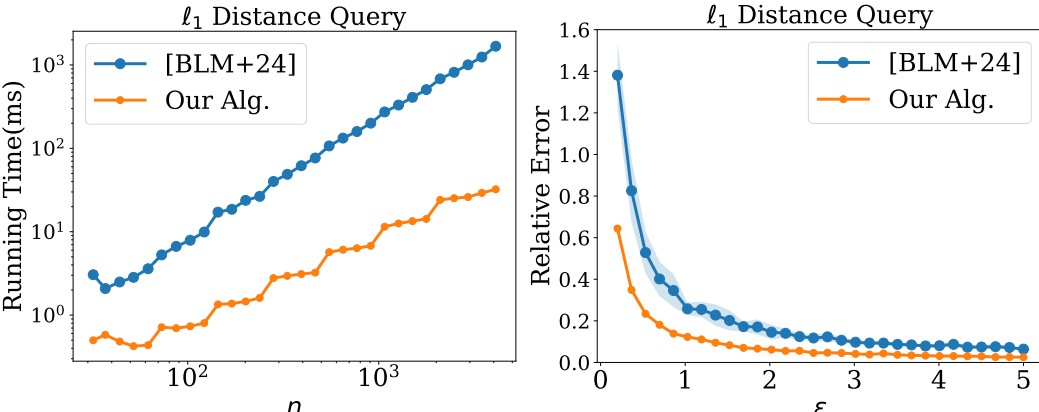

Figure 1: Running Time for Different Size $n$          Figure 2: Relative Error for Different $\epsilon$

Here we present minimally sufficient numerical results to back up our theoretical analysis of $\ell_1$ algorithm.

**Experiments Setup.** In the $\ell_1$ experiments, the objective is to compute function $f(y) = \sum_{x \in X} |x - y|$. For each experiment, we compute the average of 15 trials, with the standard deviation ($\pm 1$) shaded where appropriate.

**Computational Resource.** We conduct all experiments on a laptop with Intel i7-9750H CPU and 16 GB RAM. All codes are implemented in Python 3.12.3.

**Baseline.** We choose (Backurs et al., 2024) as our baseline and compare our algorithm with theirs on both running time and relative error. In their algorithm, we set the hyper-parameter $\alpha = 0.1$ which corresponds to the experiments settings in (Backurs et al., 2024).

**Approximation Test.** We first show that as $\epsilon$ increases, both (Backurs et al., 2024) and our algorithm approximate the true function $f$. To illustrate this, our first synthetic dataset $X$ consists of 1000 data points evenly distributed in the range $[0, 1]$, with query points also evenly distributed within the same range. In this case, the ground truth function $f$ is approximately equal to the integral $f(y) = \int_0^1 |x - y| \, dx = y^2 - y + 1/2$ for $y \in [0, 1]$. This setup allows for an easy comparison of our output to the true function. In Figure 3, we show that both the outputs of (Backurs et al., 2024) and our algorithm approximate the true function $f$ as $\epsilon$ increases, with our method achieving a faster convergence rate.

**Running Time Test.** In Figure 1, we compare the running time and relative error between (Backurs et al., 2024) and our algorithm. For running time comparison, our datasets $X$ consists of random points in $[0, 1]$ with different sizes ranging from 32 to 4096. The number of queries equals to the

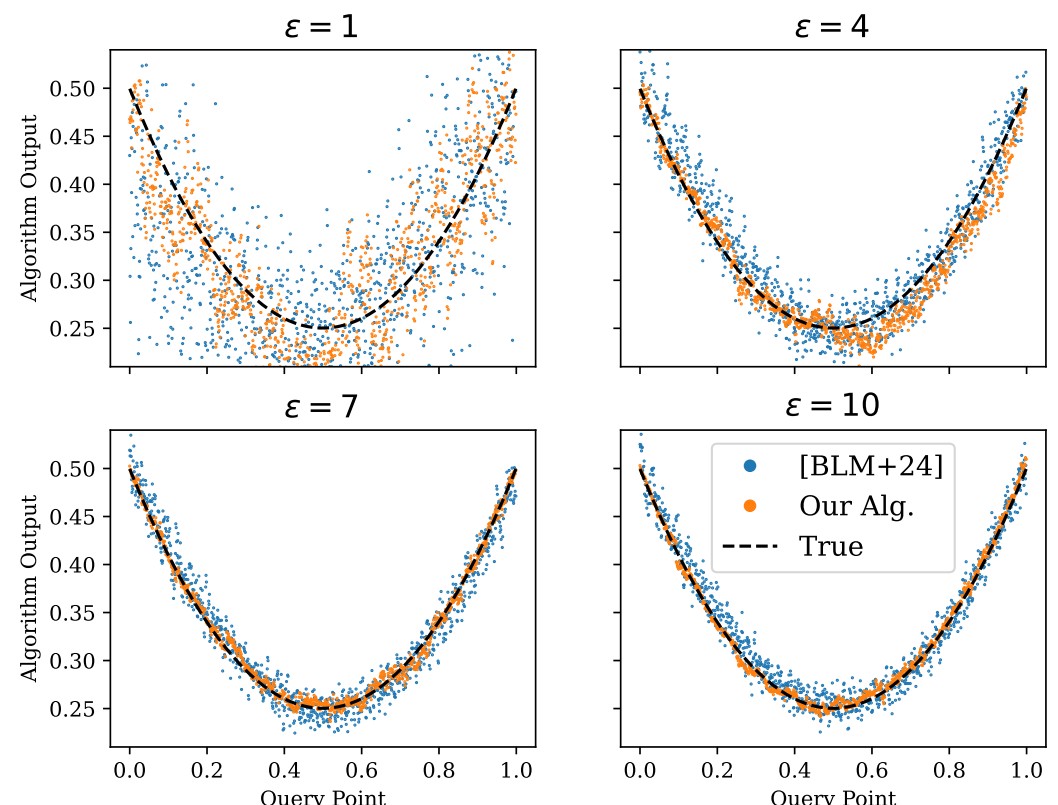

Figure 3: We benchmark our results against those of the previous best (Backurs et al., 2024). The dashed line is the ground truth. The orange and blue dots correspond to our method and (Backurs et al., 2024), respectively. While, both algorithms approximate the true $\ell_1$ distance as $\epsilon$ increases, it is clear that our method delivers more accurate and more robust predictions.

size of $|X|$. We demonstrate that our algorithm consistently has a lower running time than (Backurs et al., 2024).

**Relative Error Test.** In Figure 2, we set $|X| = 1000$ and compute relative errors. Every data point and query point is uniformly chosen from $[0, 1]$ independently. The number of queries also equals to the size of $|X|$. For every $\epsilon$ ranging from 0.2 to 5, our algorithm consistently achieves a lower relative error compared to (Backurs et al., 2024).

These numerical results align with our theory: our algorithm improves (Backurs et al., 2024) in both running time and relative error.

A.2   REAL-WORLD DATA

Here we extend our minimally sufficient numerical results to real-world data (CIFAR-10).

**Data.** CIFAR-10 is a dataset consisting of 60000 32x32 color images. We project each of them into a real vector of dimension 100 and compute distance queries for various $\epsilon$.

**Experiments Setup.** We conduct new experiments on CIFAR-10 to compare the performances between our algorithm and (Backurs et al., 2024).

In our experiments, we choose different $\epsilon \in \{0.1, 0.5, 1.0, 1.5, 2.0, 2.5, 3.0\}$. Then we construct a database containing 10000 vectors of dimension 100 and each time we process 100 queries randomly drawn from the CIFAR-10 test batch.

**Relative Error Test.** We consider three types of distance queries: $\ell_1$ (Figure 4(a)), $\ell_2$ (Figure 4(b)) and $\ell_p^p$ with $p = 3$ (Figure 4(c)). For each of them, we compute the average relative error of the

queries. Figure 4 shows that our algorithm outperforms (Backurs et al., 2024) on all of the three distance metrics.

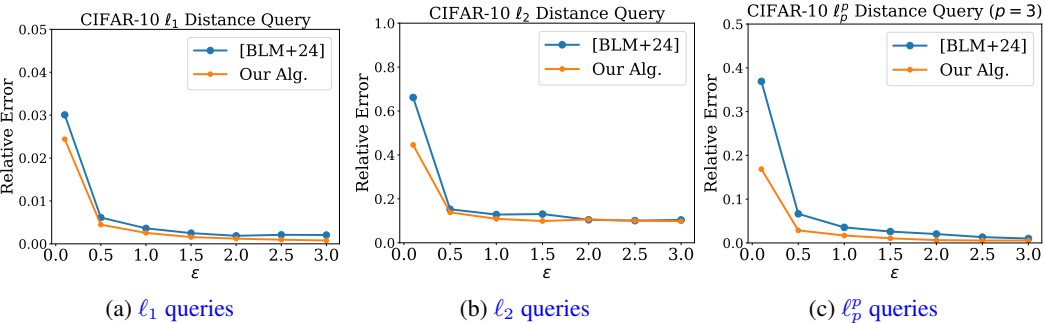

(a) $\ell_1$ queries          (b) $\ell_2$ queries          (c) $\ell_p^p$ queries

Figure 4: We conduct different types of queries on CIFAR-10. The blue line is the (Backurs et al., 2024) and the yellow line is the result of our algorithm.

## B  RELATED WORK

**Privacy, Security and Safety in Large Foundation Models.** The motivating problems for this work come from the privacy concerns of large foundation models (Lin et al., 2024; Xie et al., 2024a). In modern machine learning, Foundation models (Bommasani et al., 2021), including pretrained transformer and diffusion models, gain popularity in many AI applications due to their ability to generalize across diverse tasks with minimal fine-tuning. Pretrained transformers, such as BERT (Devlin et al., 2018), GPT (Brown et al., 2020), and Llama (Touvron et al., 2023b;a), leverage vast amounts of data to learn general-purpose, context-aware representations. Diffusion models (Peebles & Xie, 2023; Ho et al., 2020; Song & Ermon, 2019), on the other hand, excel in generative tasks, particularly in producing high-quality images and data distributions through iterative refinement (Nichol et al., 2021; Ramesh et al., 2022; Liu et al., 2024; Zhou et al., 2024a;b; Wang et al., 2024a;b). Importantly, what makes them fundamental and versatile is their flexibility for diverse downstream tasks via fine-tuning methods (Zheng et al., 2024; Ding et al., 2022; Lester et al., 2021; Liu et al., 2021; Hu et al., 2022), hence "foundation." Together, these models signify a shift towards more powerful AI systems that serve as the basis for a wide range of applications across different domains.

However, there exists a gap from productizing many these advancements due to privacy, safety, and security concerns (Qi et al., 2024; Sun et al., 2024; Li et al., 2024b; Weidinger et al., 2021). These models, due to their vast scale and extensive training on large datasets, risk exposing sensitive information and amplifying biases present in the data. Privacy issues arise when models inadvertently memorize and reproduce personal data from training sets (Carlini et al., 2023b; 2021). Safety concerns include the potential for generating harmful or misleading content, especially in applications where accuracy is paramount (Weidinger et al., 2021; OpenAI, 2023; Team et al., 2023; Anthropic, 2023; Yu et al., 2024b; Luo et al., 2024; Shen et al., 2024; Deng et al., 2023; Liu et al., 2023b;a; Shah et al., 2023; Yu et al., 2023). Security vulnerabilities also exist, as these models may be susceptible to adversarial attacks that manipulate outputs or extract proprietary information (Zeng et al., 2024; Jiang et al., 2024; Xie et al., 2024b). Addressing these challenges is essential to ensure the responsible and secure use of foundation models across various applications. In this work, we focus on the privacy of foundation models at a fundamental level by study the formalized KDE query Problem 1.1.

**Differential Privacy (DP).** Differential Privacy (DP) is a standard tool for understanding and mitigating privacy concerns in machine learning. Proposed by Dwork et al. (2006), DP offers a robust approach to protecting sensitive information. Given the fact that non-private ML models expose sensitive user data (Fredrikson et al., 2015; Carlini et al., 2019; Chen et al., 2020; Carlini et al., 2021; 2022; 2023a; Haim et al., 2022; Tramèr et al., 2022; Carlini et al., 2023b), researchers propose various methodologies. One approach involves generating synthetic datasets that closely resemble the original private data and training models on these synthetic datasets (Lin et al., 2020; Li et al., 2022; Yu et al., 2022; Yin et al., 2022; Yue et al., 2023; Lin et al., 2024). Another strategy is to use similar public examples for pre-training models (Hou et al., 2023; Yu et al., 2024a). Additionally, the

DP-SGD method has benefited from incorporating public data similar to private datasets to enhance downstream model performance (Yu et al., 2021; De et al., 2022; Yu et al., 2022; Yin et al., 2022; Li et al., 2022; Hou et al., 2023; Yu et al., 2024a; Lin et al., 2024). As highlighted by Backurs et al. (2024), the common denominator of all these works is the need to *compute similarities to a private dataset*. In this work, we study this problem at a fundamental level by formalizing it as the DP-KDE query problem (Problem 1.1).

**Kernel Density Estimation (KDE).** Kernel Density Estimation (KDE) is a key technique in statistics and machine learning. This technique converts a collection of data points into a smoothed probability distribution (Schölkopf & Smola, 2002; Shawe-Taylor & Cristianini, 2004; Hofmann et al., 2007). It is prevalent in many private applications such as crowdsourcing and location sharing (Huai et al., 2019; Cunningham et al., 2021). Although research on non-private KDE already produce efficient methods (Backurs et al., 2018; 2019; Alman et al., 2020; Charikar et al., 2020; Liang et al., 2022; Qin et al., 2022; Bakshi et al., 2022; Huang et al., 2022; Deng et al., 2022), adapting these techniques to DP remains complex (Gupta et al., 2012; Blum et al., 2013; Aldà & Rubinstein, 2017; Wagner et al., 2023; Backurs et al., 2024). The best algorithm to date is by Backurs et al. (2024). In this work, we introduce a new data structure for DP-KDE that improves upon (Backurs et al., 2024) in both privacy-utility trade-off and efficiency.

**Comparison with Function-Sanitizing Approaches for Differentially Private KDE.** Hall et al.(2013) (Hall et al., 2013) pioneer DP kernel density estimation by adding a Gaussian-process perturbation to the entire density function. This yields an $(\epsilon, \delta)$-DP sanitized function that can be queried anywhere, but it requires heavy noise and came with limited accuracy guarantees. Mirshani et al.(2019) (Mirshani et al., 2019) extend this "function release" approach using Gaussian process noise in Banach spaces, likewise achieving $(\epsilon, \delta)$-DP over the full function at a high privacy cost. Aldà and Rubinstein (2017) (Aldà & Rubinstein, 2017) instead propose the *Bernstein mechanism*, approximating the KDE with a finite polynomial basis and adding independent noise to each coefficient. This produces a published density function under (pure) DP with improved utility but requires truncating the function to a fixed basis. Smith et al.(2018) (Smith et al., 2018) apply DP to Gaussian process models of the KDE. They show that releasing an unrestricted GP-based density demands large noise, and they improve accuracy by assuming a fixed set of query points (reducing query flexibility). All these methods treat the KDE as a static function to privatize and publish up front, incurring a large one-time privacy cost or forcing constraints on queries. In contrast, our work uses a specialized $\ell_1$ kernel decomposition and a balanced tree data structure to answer KDE queries efficiently under DP. Rather than outputting an explicit density estimate, we privately pre-compute a tree that returns kernel density queries *on the fly* in sublinear time. This design concentrates the privacy budget on the queried regions and exploits the $\ell_1$ kernel's structure Consequently, our proposal yields higher accuracy and faster query responses than prior DP-KDE methods under the same privacy requirements.

**DP Tree and Matrix mechanisms.** Dwork et al. (2010a) and Chan et al. (2011) pioneer tree-based mechanisms for answering range queries under differential privacy. Their key insight is to store counts in a balanced binary tree (or equivalently use the matrix mechanism) so that each query touches only $O(\log n)$ noisy aggregates, yielding lower error than naive solutions. Recent work by Henzinger et al. (2023) further optimize these ideas to improve continual counting bounds. Our approach builds on this line of research. We also employ a balanced binary tree for fast, private query evaluation. However, we extend the classic tree mechanism by storing both counts and partial distance sums at each node, rather than just counts. This allows us to handle $\ell_1$ kernel density estimation queries efficiently: we retrieve the relevant (noisy) partial sums from each level of the tree and combine them to compute $|x - y|$ over the dataset. Thus, while our algorithm inherits the core principle of aggregating noisy statistics along a tree, it generalizes the framework to distance-weighted queries, reducing error and removing the need for an additional approximation factor.

## C   Limitations and Future Work

**Limitations.** Our technique and analysis focus *only* on *static* datasets and the $\ell_1$, $\ell_2$, and $\ell_p^p$ kernels; extending the approach to other similarity measures or dynamic/streaming data remains an open challenge. While our space and preprocessing time match existing solutions, they are still $O(nd)$ and may become restrictive in very high dimensions or for extremely large $n$. Moreover, although we reduce the query time by an $\alpha^{-1} \log n$ factor and improve approximation/error bounds,

the constants involved in the data structure could be non-negligible in practice, especially under tight privacy constraints.

**Future Work.** Our novel tree construction strategy may inspire further research on advanced data structures for privacy-preserving query processing. Future work could explore refining these constants, handling broader classes of kernels, and adapting the method to incremental or online data settings. We leave these directions for future exploration.

## D    BASIC FACTS

**Fact D.1.** *Let $X$ denote a random variable with $\mathbb{E}[X] = 0$. Then it holds*
$$\mathbb{E}[|X|] \leq (\mathrm{Var}[X])^{1/2}.$$

*Proof.* It is easy to see that $(\mathbb{E}[|X|])^2 \leq \mathbb{E}[|X|^2] = \mathbb{E}[X^2]$. Next, we show
$$(\mathbb{E}[|X|])^2 \leq \mathbb{E}[X^2] = \mathbb{E}[X^2] - (\mathbb{E}[X])^2 \leq \mathrm{Var}[X].$$
Thus, we complete the proof. □

**Fact D.2** (Union Bound). *Let $A_1, A_2, ..., A_n$ be a set of probability events. It holds*
$$\Pr\left[\bigcup_{i=1}^{n} A_i\right] \leq \sum_{i=1}^{n} \Pr[A_i].$$

*Proof.* We prove union bound by induction. For $n = 1$, $\Pr[A_1] = \Pr[A_1]$. Suppose this inequality holds true for $k$, then for $k + 1$,
$$\Pr\left[\bigcup_{i=1}^{k+1} A_i\right] = \Pr\left[\bigcup_{i=1}^{k} A_i\right] + \Pr[A_{k+1}] - \Pr\left[(\bigcup_{i=1}^{k} A_i)\bigcap A_{k+1}\right]$$
$$\leq \Pr\left[\bigcup_{i=1}^{k} A_i\right] + \Pr[A_{k+1}]$$
$$\leq (\sum_{i=1}^{k} \Pr[A_i]) + \Pr[A_{k+1}]$$
$$= \sum_{i=1}^{k+1} \Pr[A_i].$$
Thus, we complete the proof. □

## E    $\ell_2$ DISTANCE QUERY

Using the same technique in (Backurs et al., 2024), we extend our $\ell_1$ result to $\ell_2$ norm. With our faster tree data structure, we reduce the error in (Backurs et al., 2024) from $(1 + \alpha, O(\epsilon^{-1}\alpha^{-1.5}R\log^2 n))$ to $(1 + \alpha, O(\epsilon^{-1}\alpha^{-1}R\log^2 n))$. This method uses a mapping from $\ell_2$ to $\ell_1$ space that retains distances between data points with high probability.

| References | Approx. Ratio | Error | Query Time | Init time | Space |
|---|---|---|---|---|---|
| (Backurs et al., 2024) | $1 + \alpha$ | $\alpha^{-1.5}\epsilon^{-1}R\log^2 n$ | $\alpha^{-2}(d + \alpha^{-1}\log^2 n)\log n \cdot \log(1/\alpha)$ | $\alpha^{-2}nd\log n \cdot \log(1/\alpha)$ | $n(d + \alpha^{-2}\log n \log(1/\alpha))$ |
| Theorem E.2 | $1 + \alpha$ | $\alpha^{-1}\epsilon^{-1}R\log^2 n$ | $\alpha^{-2}(d + \log n)\log n$ | $\alpha^{-2}nd\log n$ | $n(d + \alpha^{-2}\log n)$ |

Table 2: Comparison of $\|x - y\|_2$ Results with Best Known Algorithm by (Backurs et al., 2024). We ignore the big-O notation in the table, for simplicity of presentation. We use $n$ to denote the number of points in dataset. We use $d$ to denote the dimension for each point in the dataset. We assume all the points are bounded by $R$. We use the $\epsilon$ to denote $\epsilon$-DP.

### E.1 ERROR AND DP GUARANTEES

**Lemma E.1** ((Matousek, 2002)). *Let $\gamma \in (0,1)$ and define $T : \mathbb{R}^d \to \mathbb{R}^k$ by*

$$T(x)_i = \frac{1}{\beta k} \sum_{j=1}^{d} Z_{ij} x_j, i \in [k],$$

*where $\beta = \sqrt{2/\pi}$ and $Z_{ij}$ are standard Gaussians. Then for every vector $x \in \mathbb{R}^d$, we have*

$$\Pr[(1-\gamma)\|x\|_2 \le \|T(x)\|_1 \le (1+\gamma)\|x\|_2] \ge 1 - e^{-c\gamma^2 k},$$

*where $c > 0$ is a constant*

**Theorem E.2.** *Let $X \subset \mathbb{R}^d$ be a private dataset of size $n$ with a bounded diameter of $R$ in $\ell_2$. For any $\alpha \in [0,1]$, there exists an $\epsilon$-DP data structure such that for any fixed query $y \in \mathbb{R}^d$, with probability $0.99$, it outputs $Z \in \mathbb{R}$ satisfying*

$$|Z - \sum_{x \in X} \|x - y\|_2| \le \alpha \sum_{x \in X} \|x - y\|_2 + O(\epsilon^{-1}\alpha^{-1} R \log^2 n).$$

*Proof.* Let $\gamma = \alpha$, and $k = O((\log n)/\alpha^2)$. Our algorithm first maps $X$ to $T(X)$ using Lemma E.1, then solves $\ell_1$ query on $T(X)$ with Algorithm 3. By triangle inequality, we have

$$|Z - \sum_{x \in X} \|x - y\|_2|$$

$$\le \underbrace{|\sum_{x \in X} \|T(x) - T(y)\|_1 - \sum_{x \in X} \|x - y\|_2|}_{\text{err}_1} + \underbrace{|Z - \sum_{x \in X} \|T(x) - T(y)\|_1|}_{\text{err}_2}.$$

Therefore, we divide the error into two parts, $\text{err}_1$ and $\text{err}_2$. $\text{err}_1$ is from mapping $T$ (Lemma E.1), $\text{err}_2$ is from Algorithm 3.

**Bound of** $\text{err}_1$. We prove with probability $0.99$,

$$|\sum_{x \in X} \|T(x) - T(y)\|_1 - \sum_{x \in X} \|x - y\|_2| \le \alpha \sum_{x \in X} \|x - y\|_2.$$

From Lemma E.1, let $\gamma = \alpha, k = c^{-1}(\log n + c')/\alpha^2$, where $c'$ is a sufficiently large constant. We have

$$\Pr[|\|x\|_2 - \|T(x)\|_1| > \alpha\|x\|_2] \le e^{-(c' + \log n)} \le 0.01 \cdot \frac{1}{n}.$$

Therefore,

$$\Pr\left[|\sum_{x \in X} \|T(x) - T(y)\|_1 - \sum_{x \in X} \|x - y\|_2| \le \alpha \sum_{x \in X} \|x - y\|_2\right]$$

$$\ge \Pr[\forall x \in X \mid |\|x\|_2 - \|T(x)\|_1| \le \alpha\|x\|_2]$$

$$\ge 1 - \sum_{x \in X} \Pr[|\|x\|_2 - \|T(x)\|_1| > \alpha\|x\|_2]$$

$$\ge 1 - n \cdot 0.01 \cdot \frac{1}{n}$$

$$= 0.99.$$

The second step follows from union bound (Fact D.2).

**Bound of** $\text{err}_2$. After mapping $T$, with high probability, each coordinate ranges from $0$ to $R' := O(R/k)$. Therefore, the additive error caused by Algorithm 3 equals

$$|Z - \sum_{x \in X} \|T(x) - T(y)\|_1| = O(\epsilon^{-1} R' k^{1.5} \log^{1.5} n)$$

$$= O(\epsilon^{-1} \frac{R\alpha^2}{\log n}(\frac{\log n}{\alpha^2})^{1.5} \log^{1.5} n)$$

$$= O(\alpha^{-1}\epsilon^{-1} R \log^2 n),$$

where the second step follows from choice of $R'$.

This completes the proof. □

### E.2 INIT AND QUERY TIME

Finally, we prove the init time and query time of our algorithm for $\ell_2$ kernel distance.

**Init Time.** For initializing, our algorithm contains two steps. First we map each data point from $d$-dimensional to $k$-dimensional. This step is a matrix multiplication and hence takes $O(ndk)$ time. Then we build a balanced binary tree. This step takes $O(nk)$ time. Therefore, given $k = (\log n)/\alpha^2$, the total init time is $O(\alpha^{-2}nd\log n)$.

**Query Time.** For each query, our algorithm also contains two steps. First we map query $y$ from $d$-dimensional to $k$-dimensional. This step is a matrix multiplication and hence takes $O(dk)$ time. Then we query $y$ on the balanced binary tree. This step takes $O(k \log n)$ time. Therefore, given $k = (\log n)/\alpha^2$, the total query time is $O(\alpha^{-2}(d + \log n)\log n)$.

## F $\ell_p^p$ DISTANCE QUERY

Suppose $p \geq 1$ is an integer. To calculate $\ell_p^p$ distance queries, we use binomial expansion techniques. Similar to $\ell_1$, we first consider the one-dimensional case.

| References | Approx. Ratio | Error | Query Time | Init time | Space |
|---|---|---|---|---|---|
| (Backurs et al., 2024) | $1 + \alpha$ | $\epsilon^{-1}\alpha^{-0.5}R^p d^{1.5}\log^{1.5} n$ | $\alpha^{-1}d\log^2 n$ | $nd$ | $nd$ |
| $\ell_p^p$ | $1$ | $\epsilon^{-1}p^{-0.5}R^p d^{1.5}\log^{1.5} n$ | $d\log n$ | $(n + pn^{1/p})d$ | $pn^{1/p}d$ |

Table 3: Comparison of $\|x - y\|_p^p$ Results with Best Known Algorithm by (Backurs et al., 2024). We ignore the big-O notation in the table, for simplicity of presentation. We use $n$ to denote the number of points in dataset. We use $d$ to denote the dimension for each point in the dataset. We assume all the points are bounded by $R$. We use the $\epsilon$ to denote $\epsilon$-DP.

**Remark F.1.** *We remark that for the small $p$ regime $p \in [1, 2]$, our result (see Table 3) match the init time and space of (Backurs et al., 2024), improve the approximation from $(1 + \alpha)$ to 1, reduce the error by a factor of $\alpha^{-0.5}$, and reduce the query time by a factor of $\alpha^{-1}\log n$.*

We first extend our key observation Lemma 3.1 with binomial expansion.

**Lemma F.2.** *For a collection of numbers $\{x_1, x_2, \cdots, x_n\} \subset \mathbb{R}$ and a number $y \in \mathbb{R}$. We define two sets*

$$S_+ := \{k \in [n] \ : \ x_k > y\}$$
$$S_- := \{k \in [n] \ : \ x_k < y\},$$

*It holds*

$$\sum_{k=1}^{n} |x_k - y|^p = \sum_{j=0}^{p} \binom{p}{j} y^{p-j}((-1)^{p-j}\sum_{k \in S_+} x_k^j + (-1)^j \sum_{k \in S_-} x_k^j),$$

*where $\binom{p}{j}$ denotes the binomial coefficient that $\binom{p}{j} = \frac{p!}{j!(p-j)!}$.*

*Proof.* We show that

$$\sum_{k=1}^{n} |x_k - y|^p = \sum_{x_k \in S_+} (x_k - y)^p + \sum_{x_k \in S_-} (y - x_k)^p$$

$$= (\sum_{x_k \in S_+} \sum_{j=0}^{p} (-1)^{p-j}\binom{p}{j}x_k^j y^{p-j}) + (\sum_{x_k \in S_-} \sum_{j=0}^{p} (-1)^j \binom{p}{j}x_k^j y^{p-j})$$

$$= \sum_{j=0}^{p} (\binom{p}{j}(-1)^{p-j}y^{p-j}\sum_{k \in S_+} x_k^j) + \sum_{j=0}^{p} (\binom{p}{j}(-1)^j y^{p-j}\sum_{k \in S_-} x_k^j)$$

$$= \sum_{j=0}^{p} \binom{p}{j} y^{p-j} ((-1)^{p-j} \sum_{k \in S_+} x_k^j + (-1)^j \sum_{k \in S_-} x_k^j).$$

Thus, we complete the proof. □

Lemma F.2 presents a decomposition of the final answer. To calculate the final answer, on each node, we store $\sum_{x_k \in I} x_k^q$ for all $q \in \mathbb{N}, 0 \le q \le p$. $I$ denotes the interval that the node maintains. For each query, we define $s_{\text{left},q}$ and $s_{\text{right},q}$ as $\sum_{k \in S_-} x_k^q$ and $\sum_{k \in S_+} x_k^q$ respectively. Then we calculate them and construct the final answer.

---

**Algorithm 4** $\ell_p^p$ Distance Query

---

1: **data structure** $\ell_p^p$ FASTERTREE                                            ▷ Theorem F.3
2: **members**
3:    $T := \{s_{l,j,q}\}$                ▷ $s_{l,j,q}$ represents $\sum_{x_k \in [(j-1) \cdot \frac{R}{2^{l-1}}, j \cdot \frac{R}{2^{l-1}})} x_k^q$ as defined above.
4: **end members**
5:
6: **procedure** INIT$(X, n, \epsilon, L)$
7:    **for** each $x_k$ in $X$ **do**
8:        find the leaf node $j$ of layer $L$ that $x_k$ is in the interval $I_{L,j} = [(j-1) \cdot \frac{R}{2^{L-1}}, j \cdot \frac{R}{2^{L-1}})$
9:        **for** $q$ from 0 to $p$ **do**
10:            $s_{L,j,q} \leftarrow s_{L,j,q} + x_k^q$
11:        **end for**
12:    **end for**
13:    **for** each layer $l$ from $L-1$ to 1 **do**
14:        **for** each node $j$ in layer $l$ **do**
15:            **for** $q$ from 0 to $p$ **do**
16:                $s_{l,j,q} \leftarrow s_{l+1,2j-1,q} + s_{l+1,2j,q}$
17:            **end for**
18:        **end for**
19:    **end for**
20:    **for** $q$ from 0 to $p$ **do**
21:        Add independent noises drawn from $\mathsf{Laplace}(pLR^q/\epsilon)$ to each $s_{l,j,q}$
22:    **end for**
23:    $T \leftarrow \{s\}$
24: **end procedure**
25:
26: **procedure** QUERY$(y \in \mathbb{R})$
27:    **for** $q$ from 0 to $p$ **do**
28:        $s_{\text{left},q}, s_{\text{right},q} \leftarrow 0$
29:    **end for**                                                          ▷ As previously defined
30:    **for** each layer $l$ from 2 to $L$ **do**
31:        find the node $j$ of layer $l$ that $y$ is in the interval $I_{l,j} = [(j-1) \cdot \frac{R}{2^{l-1}}, j \cdot \frac{R}{2^{l-1}})$
32:        **for** $q$ from 0 to $p$ **do**
33:            **if** $j \mod 2 = 0$ **then**                    ▷ if node $j$ is the right child of its parent
34:                $s_{\text{left},q} \leftarrow s_{\text{left},q} + s_{l,j-1,q}$
35:            **else**                                            ▷ if node $j$ is the left child of its parent
36:                $s_{\text{right},q} \leftarrow s_{\text{right},q} + s_{l,j+1,q}$
37:            **end if**
38:        **end for**
39:    **end for**
40:    **return** $\sum_{q=0}^{p} \binom{p}{q} y^{p-q} ((-1)^{p-q} s_{\text{right},q} + (-1)^j s_{\text{left},q})$      ▷ Lemma F.2
41: **end procedure**
42: **end data structure**

---

**Theorem F.3** (1-Dimensional $\ell_p^p$ DP Distance Query). *Given a dataset $X \subset \mathbb{R}$ with $|X| = n$, there is an algorithm that uses $O(pn^{1/p})$ space to build a data structure (Algorithm 4) which supports the following operations*

- **INIT**$(X, \epsilon)$. *It takes dataset $X$ and privacy parameter $\epsilon$ as input and spends $O(n + pn^{1/p})$ time to build a data-structure.*

- **QUERY**$(y \in \mathbb{R})$. *It takes $y$ as input and spends $O(\log n)$ time to output a scalar $A$ such that $\mathbb{E}[|A - A'|] \leq O(p^{-0.5} \epsilon^{-1} R^p \log^{1.5} n)$.*

*Furthermore, the data structure is $\epsilon$-DP.*

*Proof.* We set the total layers $L = (\log n)/p$.

**Init Time.** The total number of nodes on the tree is $2^{L+1} - 1 = O(n^{1/p})$. Each node stores $p$ values, so there are $O(pn^{1/p})$ values stored on the tree. Plus the time of iterating all data points, initializing these values takes $O(n + pn^{1/p})$ time.

**Query Time.** Each query iterate through all layers. On each layer it takes $O(p)$ time to calculate $s_{\text{left},q}$ and $s_{\text{right},q}$. There are $(\log n)/p$ layers, so the total query time is $O(\log n)$.

**Privacy Guarantees.** Similar to Lemma 3.5, we consider each $q$ separately. For $q$, let $F_q(X) : [0, R)^n \to \mathbb{R}^{2^{L+1}-1}$ be the mapping from dataset $X$ to $s_{l,j,q}$ of all $l, j$. Next we prove each $F_q(X)$ is $(\epsilon/p)$-DP.

For $F_q(X)$, when changing each data point, at most one node changes at each layer. Each node changes by at most $O(R^q)$. Therefore, the sensitivity is $O(LR^q)$. Adding coordinate-wise Laplace noise with magnitude $\eta = O(pLR^q/\epsilon)$ suffices to ensure $(\epsilon/p)$-DP using the standard Laplace mechanism.

By Lemma 2.2, the differential privacy parameter $\epsilon$ of the tree $T := (F_q(X) : 0 \leq q \leq p)$ equals $p \cdot \epsilon/p = \epsilon$. This completes the proof.

**Error Guarantees.** Similar to Lemma 3.6, the additive error consists of two parts.

The first part is from the data in the leaf node which contains query $y$. The error is

$$\sum_{x_k \in [(j-1) \cdot R/2^L, j \cdot R/2^L)} |x_k - y|^p \leq n \cdot \left(\frac{R}{2^L}\right)^p.$$

When $L = (\log n)/p$, this error is $O(R^p)$.

The second part is the Laplace noise. We first show that

$$\mathbb{E}[|\sum_{i=1}^{L} \mathsf{Laplace}(\lambda)|] \leq \sqrt{\mathrm{Var}[\sum_{i=1}^{L} \mathsf{Laplace}(\lambda)]}$$

$$\leq \sqrt{L \cdot \mathrm{Var}[\mathsf{Laplace}(\lambda)]}$$

$$= \sqrt{2L\lambda^2}$$

$$= \sqrt{2L}\lambda,$$

where the first step follows from Fact D.1, the third step follows from $\mathrm{Var}[\mathsf{Laplace}(x)] = 2x^2$.

Replacing $\lambda = pR^pL/\epsilon$, we have

$$\mathbb{E}[|\sum_{i=1}^{L} \mathsf{Laplace}(pR^pL/\epsilon)|] \leq \sqrt{2}pR^pL^{1.5}/\epsilon. \tag{F.1}$$

Then we bound the error with this inequality:

$$\mathbb{E}[|A - A'|] \leq \mathbb{E}[|\sum_{q=0}^{p} \binom{p}{q} y^{p-q} \sum_{i=1}^{L} ((-1)^{p-j} \mathsf{Laplace}(pR^qL/\epsilon) + (-1)^j \mathsf{Laplace}(pR^qL/\epsilon))|]$$

$$\leq \sum_{q=0}^{p} \binom{p}{q} y^{p-q} \mathbb{E}[|\sum_{i=1}^{L} (\mathsf{Laplace}(pR^qL/\epsilon) + \mathsf{Laplace}(pR^qL/\epsilon))|]$$

$$= \sum_{q=0}^{p} \binom{p}{q} y^{p-q} \cdot 2pR^q L^{1.5}$$

$$= 2pL^{1.5} \sum_{q=0}^{p} \binom{p}{q} y^{p-q} R^q$$

$$= 2pL^{1.5}(y + R)^p$$

$$= O(p^{-0.5} R^p \log^{1.5} n),$$

where the third step follows from Eq. (F.1). The last step is from $L = (\log n)/p, y \in [0, R)$. □

For high dimensional $\ell_p^p$ queries, we use the same procedures in Section 3.6. We build $d$ independent data structures for each dimension following Algorithm 3. We also compare the privacy and error guarantees with those in (Backurs et al., 2024) in Table 3.

**Theorem F.4** ($d$-Dimensional $\ell_p^p$ DP Distance Query). *Given a dataset $X \subset \mathbb{R}^d$ with $|X| = n$, there is an algorithm that uses $O(pn^{1/p}d)$ space to build a data-structure which supports the following operations*

- **INIT**$(X, \epsilon)$. *It takes dataset $X$ and privacy parameter $\epsilon$ as input and spends $O(nd + pn^{1/p}d)$ time to build a data-structure.*

- **QUERY**$(y \in \mathbb{R}^d)$. *It takes $y$ as input and spends $O(d \log n)$ time to output a scalar $A$ such that $\mathbb{E}[|A - A'|] \leq O(p^{-0.5}\epsilon^{-1}R^p d^{1.5} \log^{1.5} n)$.*

*Furthermore, the data structure is $\epsilon$-DP.*

*Proof.* We omit the proof as it follows from Section 3.6 in a straightforward manner. □

## G DIAGRAM EXAMPLES SHOWING THE DIFFERENCE BETWEEN (BACKURS ET AL., 2024)'S AND OURS ALGORITHMS

Here we provide visualization examples for Section 2.2 and Section 2.3 to contrast our proposal with that of (Backurs et al., 2024). To better understand our idea, let us consider the non-DP version, which has no noise. Moreover, we simplify our tree structure. By "simplify", we mean that, we drop some information from our trees even before adding any DP noise.

Under this setting, here we provide a simple way to show the difference between their algorithm and ours. Suppose the data is [1, 2, 3, 4].

- **(Backurs et al., 2024)'s tree** is a classical binary tree, as shown in Figure 5. Each leaf stores the original value, and each intermediate node stores the sum of its two children.
- **Our tree** is a prefix-sum tree, as shown in Figure 6. Each intermediate node stores the prefix-sum immediately to the left of all its descendant leaves. For example, node $c$ (value 3) stores the value of node $f$. Each leaf node stores the prefix-sum from the leftmost leaf node up to itself. For another example, leaf node $g$ stores the sum of all values to its left.

For details on prefix-sum trees, we refer readers to the following standard references. The first is by CMU professor Guy Blelloch's lecture notes, "Prefix Sums and Their Applications" (Blelloch, 1990). The second is the textbook "Programming Massively Parallel Processors: A Hands-On Approach" (Kirk & Hwu, 2016). The last is the classic algorithms textbook, "Introduction to Algorithms" (3rd Ed.) (Cormen et al., 2009).

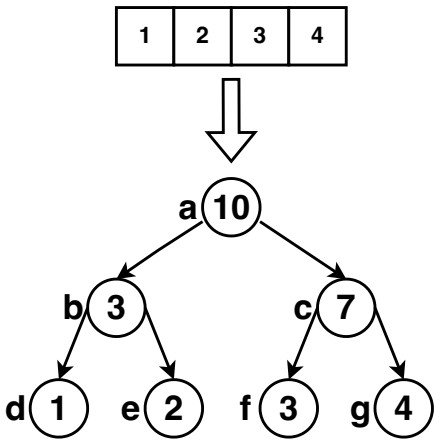

Figure 5: Example of binary tree

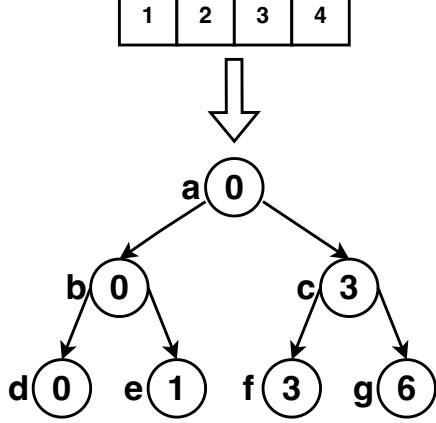

Figure 6: Example of prefix-sum tree

