# OpenReview forum: "On Differential Private $\ell_1$, $\ell_2$ and $\ell_p^p$ Distance Queries"
_ICLR.cc/2026/Conference — ICLR 2026 Conference Desk Rejected Submission_

### Official Review · Reviewer_zMT4 · 2025-10-20

**Soundness:** 3
**Presentation:** 2
**Contribution:** 3
**Rating:** 4
**Confidence:** 2

**Summary:**

This paper introduces a refined differentially private data structure for kernel density estimation with $l_1$, $l_2$, and $l_p^p$ distance kernels. The proposed method theoretically improves upon the current state-of-the-art (Backurs et al., ICLR 2024) in query time, approximation ratio, and additive error. While the theoretical contribution is significant, the experimental validation is somewhat limited, and the writing contains several typographical errors that need addressing.

**Strengths:**

1. Theoretical Contribution: This paper makes a clear and valuable theoretical advance. The novel balanced binary tree structure, which stores both distance sums and counts, successfully reduces the query time by a factor of $\alpha^{-1} \log n$ and improves the error bounds by eliminating the multiplicative $(1+\alpha)$ approximation and reducing the additive error dependence on $\alpha$.
2. Technical Novelty: The key observation (Lemma 3.1) that decomposes the $l_1$ distance into four components is elegant and drives the efficiency gains. This core idea is cleanly extended to $l_2$ and $l_p^p$ distances in the appendices, demonstrating the generality of the approach.
3. Rigorous Analysis: The paper provides a thorough theoretical analysis, including proofs for initialization time, query time, differential privacy guarantees (using the Laplace mechanism), and error bounds. The analysis appears sound and follows established DP composition theorems.

**Weaknesses:**

1. Limited Experimental Evaluation: The experiments are conducted only for the 1-dimensional $l_1$ case, leaving the performance on higher-dimensional data and for $l_2$/$l_p^p$ kernels invalidated. The scale of the experiments is relatively small (n ≤ 4096), and it is unclear how the method performs on large-scale, real-world datasets. The figures (e.g., Figure 3) lack clear axis labels and detailed captions, making them difficult to interpret and the results hard to reproduce.
2. Writing and Presentation: There are several typos and unclear statements throughout the text. The error analysis in the proof of Lemma 3.6 contains confusing formula references. The range and impact of the parameter p in the $l_p^p$ section could be explained more clearly. Some proofs, particularly for the high-dimensional extensions, are somewhat terse and could benefit from more detailed exposition.
3. Practical Limitations: The proposed data structure is designed for static datasets and does not support dynamic updates. The space complexity of $O(nd)$ could become a bottleneck for very high-dimensional data.

**Questions:**

1. Experimental Evaluation: Can the authors provide experimental results for higher dimensions (d > 1) and for the $\ell_2$ and $\ell_p^p$ kernels to comprehensively validate the theoretical claims?  Could the algorithm be compared against a wider range of baseline methods, such as DP-KDE methods based on hashing or sampling? Furthermore, the number of baselines compared in this paper is insufficient. It should include comparisons with more high-level methods and the design of additional experimental schemes, as the current experimental content appears rather limited.
2. References: The references are not sufficiently up-to-date. The authors should pay attention to more high-quality literature published since 2025. Additionally, this paper has cited an excessive number of arXiv preprints. Is this appropriate? The authors should thoroughly review the related work and references. Moreover, the writing styles of the abstract and conclusion in this paper do not seem to conform to standard conventions. The authors should further enhance their presentation in this regard.
3. Theoretical Details: For $l_p^p$ queries, how does the value of p impact practical performance? Does the method support non-integer p? Has the authors considered using other noise mechanisms (e.g., Gaussian noise) to achieve $(\epsilon, \delta)$-DP, and how would that affect the error bounds?
4. Writing Improvements: It is recommended to number all Algorithms/Equations and reference them clearly in the main text. All figures should be improved with proper axis labels, legends, and descriptive captions. A thorough proofreading is necessary to correct typographical errors and improve the clarity of explanations.
5. The supplementary code material is incomplete and lacks explanations for reproduction. The authors need to provide complete and credible reproducible code.

---

> ### Author Response · Authors · 2025-11-25
> **Rebuttal 1**
>
> ### We thank the reviewer for the detailed review. In response, we have addressed all concerns and questions in the following replies and revisions in the updated draft. All changes from the originally submitted version are highlighted in blue in the revised PDF.
>
> > **W1.** Limited Experimental Evaluation....
>
> Thanks for your suggestions. In response, we have added new experiments on real-world data for all $\ell_1,\ell_2,\ell_p$ kernels. Please see our newly added Appendix A.1 for details. We remark that, these new results also align well with our theory and algorithm.
>
> > **W2.** Writing and Presentation: There are several typos and unclear statements throughout the text. The error analysis in the proof of Lemma 3.6 contains confusing formula references. The range and impact of the parameter p in the $l_p^p$ section could be explained more clearly. Some proofs, particularly for the high-dimensional extensions, are somewhat terse and could benefit from more detailed exposition.
>
> Thanks for the suggestion. We have revised the draft accordingly.
>
>
> Please let us know if there any other place that we can further clarify.
>
> > **W3.** Practical Limitations: The proposed data structure is designed for static datasets and does not support dynamic updates. The space complexity of $O(nd)$ could become a bottleneck for very high-dimensional data.
>
> Thanks for the comment. We respectfully remind the reviewer that, this is beyond the current scope of this paper. Please see our abstract and intro for the precise problem definition.
>
> We also want to remind the reviewer that, our proposed algorithm (and theory) is already SOTA. Extending to scalable high-dim version is beyond the scope of this paper (yet will be interesting and important future work).
>
> > **Q1.** Experimental Evaluation: Can the authors provide experimental results for higher dimensions (d > 1) and for the $\ell_2$ and $\ell_p^p$ kernels to comprehensively validate the theoretical claims? Could the algorithm be compared against a wider range of baseline methods, such as DP-KDE methods based on hashing or sampling? Furthermore, the number of baselines compared in this paper is insufficient. It should include comparisons with more high-level methods and the design of additional experimental schemes, as the current experimental content appears rather limited.
>
> In response, we remind the reviewer, our results and algorithm already improve upon the previous best algorithm [BLM+24] in the problem considered. Therefore, benchmarking against [BLM+24] already subsumes other baselines.
>
> As mentioned in above W1, we invite the reviewer to check our new Appendix A.1 for new numerical results on real-world data for all $\ell_1,\ell_2,\ell_p$ kernels.
>
>
>  > **Q2-1.** References...
>
> Thanks for your comments. We recently updated our references right before submission, and we believe we have done our due diligence in covering the relevant literature. That said, we remain open to any specific suggestions from the reviewer.
>
> Regarding arXiv citations, you are absolutely correct. We have now updated most arXiv entries to their published venue versions wherever available.
>
> > **Q2-2.** Moreover, the writing styles of the abstract and conclusion in this paper do not seem to conform to standard conventions. The authors should further enhance their presentation in this regard.
>
> We respectfully remind the reviewer, our abstract follows the standard format commonly used in theory-oriented papers (problem setup -> prior result -> our contributions -> technical insights). This structure is the standard convention in TCS, mathematics, and ML theory papers.
>
> >  **Q3-1.** Theoretical Details: For $l_p^p$ queries, how does the value of p impact practical performance? Does the method support non-integer p?
>
>
> In response to the choice of $p$, In our paper, we prove the performance of our algorithm for all integer values of $p>1$. However, the choice of $p$ in practice depends on the specific application context, which is outside the scope of our study. The impact of $p$ on performance can vary depending on the use case.
>
> > **Q3-2.** Has the authors considered using other noise mechanisms (e.g., Gaussian noise) to achieve $(\epsilon, \delta)$-DP, and how would that affect the error bounds?
>
> Thanks for the question. We value the opportunity to clarify.
>
> In response, we use Laplace noise because it gives a simple and clean pure $\epsilon-$DP guarantee. This is simpler in form. Our data structure can also use Gaussian noise. With advanced composition, this yields an  $(\epsilon,\delta)-$DP guarantee with a different error bound.

---

> > ### Comment · Reviewer_zMT4 · 2025-11-26
> >
> > I do have quite a few doubts about this paper, yet I notice that other reviewers seem to have few questions (very limited comments). Some of the doubts I raised have not been addressed by the authors yet. For instance, the experimental data volumes used in this paper are relatively small (how can it be demonstrated that the method remains effective with large-scale data?). Also, is there really no state-of-the-art method in this field over the past 2025 year? And what about the scalability of this proposed scheme? Although the authors claim that these issues are beyond the scope of this study, I expect to see the authors' thoughts and discussions on these questions, as well as additional experiments and analyses to substantiate the superiority of their method. I believe that a high-caliber paper should feature more in-depth discussions and more comprehensive experiments/demonstrations. By reading relevant papers, I have gained a deeper understanding of this field. I will maintain my current score unless the authors provide more supporting evidence. Additionally, I encourage other reviewers to join the discussion. If, after thorough deliberation, this paper is indeed deemed to be of top-tier quality, then my stance may be reconsidered.

---

> ### Author Response · Authors · 2025-11-25
> **Rebuttal 2**
>
> > **Q4.** Writing Improvements: It is recommended to number all Algorithms/Equations and reference them clearly in the main text. All figures should be improved with proper axis labels, legends, and descriptive captions. A thorough proofreading is necessary to correct typographical errors and improve the clarity of explanations.
>
> Thanks for the suggestion. We have revise the caption of exp figures accordingly. We also conducted another round of proofreading.
>
> As for numbering the equations, we respectfully remind the reviewer that it is standard convention in TCS, ML theory, mathematics, and statistics to leave equations unlabeled when they are not referenced later. Thank you for the suggestion, though.
>
> > **Q5.** The supplementary code material is incomplete and lacks explanations for reproduction. The authors need to provide complete and credible reproducible code.
>
> Thank you for the comment. However, we respectfully remind the reviewer that, our submitted code is a self-contained jupyter notebook that anyone can execute from top to bottom to reproduce results in the paper.  We are not sure what further explanation is required for this standard layout, but we are open to concrete suggestions.

---

### Official Review · Reviewer_7WQm · 2025-10-28

**Soundness:** 3
**Presentation:** 3
**Contribution:** 3
**Rating:** 8
**Confidence:** 4

**Summary:**

This paper proposes a refined differentially private (DP) data structure for kernel density estimation (KDE) with multiple distance kernels. The theoretical results significantly improve the best-known results in terms of query efficiency and approximation factor, while reducing additive errors. Numerical experiments also demonstrate the effectiveness of the proposed method in a proof-of-concept manner.

**Strengths:**

**S1.** The theoretical results improve upon the best-known results and make clear novel contributions.

**S2.** The proposed methods show good technical quality.

**S3.** The paper is well-organized, and the presentation is generally clear and easy to follow.

**S4.** The methods and theoretical results present great significance in the domain of theoretical ML.

**Weaknesses:**

**W1.** There are some inconsistencies between $O(\cdot)$ and $\cdot$. The authors should double-check whether an expression indicates a specific number or a magnitude for complexity.

**W2.** The experiments are somehow insufficient. The authors can conduct additional experiments on real-world datasets or with other distance kernels to further validate the effectiveness of the proposed methods.

**Questions:**

See Weaknesses.

---

> ### Author Response · Authors · 2025-11-25
> **Rebuttal 1**
>
> ### We thank the reviewer for the detailed review. In response, we have addressed all concerns and questions in the following replies and revisions in the updated draft. All changes from the originally submitted version are highlighted in blue in the revised PDF.
>
>
> > **W1.** There are some inconsistencies between  and . The authors should double-check whether an expression indicates a specific number or a magnitude for complexity.
>
> Thanks for pointing this out. We have revised accordingly and conducted another round of proofreading. Please let us know if there’s any left. We are more than happy to further revise.
>
>
> > **W2.** The experiments are somehow insufficient. The authors can conduct additional experiments on real-world datasets or with other distance kernels to further validate the effectiveness of the proposed methods.
>
> Thanks for the comment. In response, we have added the new experiments on real-world data (cifar10) in our latest revision (see page 18 section A.2 in our revised pdf). These new results further confirm our theory and algorithm.

---

### Official Review · Reviewer_wUxb · 2025-10-30

**Soundness:** 3
**Presentation:** 3
**Contribution:** 3
**Rating:** 8
**Confidence:** 3

**Summary:**

Let $R > 0$, and let $X \subseteq [0, R)^d$ be a dataset of size $|X| = n$.
The paper studies the problem of constructing an $\epsilon$-differentially private data structure that can answer queries of the form
$$
    \sum_{x \in X} \Vert x - y \Vert_1 = \sum_{i = 1}^d \sum_{x \in X} \vert x_i - y_i \vert,
$$
where $y \in [0, R)^d$ is a query vector.

To address this, the paper reduces the problem to designing $\epsilon$-differentially private data structures that estimate $\sum_{x \in X} \vert x_i - y_i \vert$ for each coordinate $i \in [d]$, since the overall $\ell_1$ distance is simply the sum of these estimates.
The key observation is that
$$
    \sum_{x \in X} \vert x_i - y_i \vert
        = y_i \cdot |\{ x \in X : x_i < y_i \}| - \sum_{x \in X : x_i < y_i} x_i
          + \sum_{x \in X : x_i > y_i} x_i - y_i \cdot |\{ x \in X : x_i > y_i \}|.
$$
Hence, it suffices to construct data structures that estimate:
1. $|\{ x \in X : x_i < y_i \}|$ and $|\{ x \in X : x_i > y_i \}|$, and
2. $\sum_{x \in X : x_i < y_i} x_i$ and $\sum_{x \in X : x_i > y_i} x_i$.

The data structure is built as follows:
1. Partition $[0, R)$ into $n$ equal-sized intervals, each of length $R/n$. For simplicity, assume $n$ is a power of $2$. Each $x \in X$ is assigned to the interval it belongs to.
2. Treat each interval as a leaf node and construct a complete binary tree over these nodes.
3. For each node, maintain two quantities:
   - the sum of all $x \in X$ belonging to the leaf nodes covered by this node, and
   - the count of such $x$.
4. Add Laplace noise to each node to ensure differential privacy.

Given a query $y$, let $I_y$ denote the leaf node containing $y$, and let $X_{I_y}$ be the subset of $X$ that falls in $I_y$.
Using the binary tree, one can compute a noisy estimate of $\sum_{x \in X \setminus X_{I_y}} |x_i - y_i|$ in $O(\log n)$ time.
For the leaf node error, we have $\sum_{x \in X_{I_y}} |x_i - y_i| \le n \cdot (R/n) = R$.

This approach removes the multiplicative error in Backurs et al. (2024) while also reducing both the additive error and the query time.

The paper further extends the results to the $\ell_2$ and more general $\ell_p$ norms, although the improvements in these cases are less significant than in the $\ell_1$ case.

**Strengths:**

1. The paper achieves state-of-the-art error bounds and query times.

2. The manuscript is well written, with clearly presented ideas and carefully chosen notations.

3. The proposed algorithm leverages the structure of the $\ell_1$ distance in a clever way, leading to a simple and elegant design.

**Weaknesses:**

I did not identify any significant drawbacks in the paper.

**Questions:**

Typo (lines 217–218). The additive error is stated incorrectly. It should be
$$
O\big(\epsilon^{-1} R \log^{3/2} n\big),
$$
instead of the currently printed expression.

---

> ### Author Response · Authors · 2025-11-24
> **Thanks for your endorsement**
>
> We thank the reviewer for the strong endorsement. The identified typo will be corrected in our next update.

---

### Official Review · Reviewer_VCiU · 2025-11-14

**Soundness:** 3
**Presentation:** 4
**Contribution:** 3
**Rating:** 8
**Confidence:** 3

**Summary:**

This paper is about the (very important) problem of differentially-private kernel density estimation, offering significant improvements over the previous SOTA algorithm (Backurs 2024). Formally, the problem is to compute the following sum.

$$\sum_{x \in X} \|x - y\|_1$$

The algorithm of Backurs et. al. considers each dimension separately, reducing the d-dimensional problem to d instances of the 1-dimensional problem. For each dimension $i$, the algorithm (1) discretizes each entry from the dataset $X$ to a multiple of $R / N$, (2) counts the number of values that map to each of the $N$ discretization points, and (3) constructs a binary interval tree over the discretization points, where each node contains the sum of its childrens' counts. To query the structure, we issue interval-count queries around $y_i$ with geometrically increasing interval sizes. $\log(N)/\alpha$ interval queries are needed for the approximation (where $\alpha$ is the multiplicative approximation error) and each one has complexity $\log(N)$, which determines the overall time complexity.

In this paper, the authors modify the tree representation of Backurs et. al. in a clever way. The authors show that one can decompose the L1 distance into two sums that can be pre-computed and two sums that are query-dependent. By modifying the tree to store the pre-computed sums,
Privacy is then achieved using methods similar to prior work. The really interesting part is that this alternative representation allows us to use a different interval strategy (linear intervals that subdivide $[0,1]$ rather than geometric intervals that surround $y_i$).

The result is an algorithm that improves on both the query time and the approximation error.

**Strengths:**

1. The paper contains the best known algorithm for DP distance queries, an important problem and active area of research.
2. The arguments of the paper are presented in a way that is very easy to follow. I particularly appreciated Section 3.1 - the argument about Lemma 3.1 made the whole strategy clear.
3. The algorithm is intuitive and elegant.

**Weaknesses:**

This is a solid paper. While it could be improved by a stronger empirical evaluation (the Backurs paper had results for KDE as well as for distance queries, and unless I am mistaken there are not KDE experiments here).

My other request would be to show a diagram with your vs. their interval strategy (and perhaps a motivating example showing why the extra intervals are not necessary). It took me some time to understand how you were able to reduce the number of calls to the interval tree.

**Questions:**

Is your distance decomposition necessary to be able to use your linear intervals? I guess this is probably true, since they would fail with the old algorithm when $\|x - y\| = 0$. I suppose the reason why it works is because your strategy multiplies by $y$ after doing the interval count?

---

> ### Author Response · Authors · 2025-11-25
> **Rebuttal 1**
>
> ### Thanks for your detailed review. We have addressed all your concerns in below. We also made corresponding revisions in our latest pdf. All changes from the originally submitted version are highlighted in blue in the revised PDF.
>
> > **Weakness 1.** While it could be improved by a stronger empirical evaluation (the Backurs paper had results for KDE as well as for distance queries, and unless I am mistaken there are not KDE experiments here).
>
> Thanks for the suggestion! We have added new experiments on real-world data in Appendix A.2 of our latest revision. These new results also align well with out theory.
>
> > **Weakness 2.** My other request would be to show a diagram with your vs. their interval strategy (and perhaps a motivating example showing why the extra intervals are not necessary). It took me some time to understand how you were able to reduce the number of calls to the interval tree.
>
> Thanks for the suggestion. We agree the clarity of our initial submission is not optimal.
>
> In response, we added a diagram to show the difference between our key idea and previous work’s key idea in Appendix G (see page 26, 27). We will put it back to the main text if space permit in the final version.
>
> To better understand our idea, let us consider the non-DP version, which has no noise. Note that our trees have more information, here we did some simplification for our tree structure even without adding the DP noise version. Here we can provide a simple way to show the difference between their algorithm and ours.
>
> Suppose the data is [1, 2, 3, 4], then their tree is a classical binary tree.
>
>               (a,10)
>        (b,3)         (c,7)
>     (d,1) (e,2)   (f,3) (g,4)
>
> Here each leaf node stores the original value. Each intermediate node is storing the summation of its two children.
>
>
> Suppose the data is [1, 2, 3, 4] as well, our tree is called prefix-sum tree
>
>                (a,0)
>        （b,0)          (c,3)
>      (d,0) (e,1)   (f,3) (g,6)
>
> The intermediate node stores the prefix-sum immediately to the left of all its descendant leaves. For example, node c (value 3) stores the value of the node f. Each leaf node stores the value of all the pre-sum (from the leftmost leaf node to itself).
>
> For details of prefix-sum tree, we refer the reviewers to the following textbooks. The first one is a lecture notes by CMU professor Guy Blelloch, “Prefix sums and their applications.” [1]. The second one is a well-known textbook in the field of parallel algorithms, “Programming massively parallel processors: a hands-on approach.” [2]. The last one is the most popular algorithm textbook. “Introduction to algorithms (3rd Ed).” [3].
>
>
> [1] Guy E. Blelloch. “Prefix sums and their applications.” 1990. https://www.cs.cmu.edu/~guyb/papers/Ble93.pdf
>
> [2] David B. Kirk, and W. Hwu Wen-Mei. “Programming massively parallel processors: a hands-on approach.” Morgan kaufmann, 2016.
>
> [3] Thomas H. Cormen, Charles E. Leiserson, Ronald L. Rivest, and Clifford Stein. “Introduction to algorithms (3rd Ed).” MIT press, 2009.
>
>
>
> > **Question 1.** Is your distance decomposition necessary to be able to use your linear intervals? I guess this is probably true, since they would fail with the old algorithm when $|x - y| = 0$. I suppose the reason why it works is because your strategy multiplies by $y$ after doing the interval count?
>
> You are absolutely correct. Please also see above discussion for illustrating examples.

---

### Note · Program_Chairs · 2026-01-17
**Submission Desk Rejected by Program Chairs**

The following references in this submission do not refer to real documents and/or have major errors in bibliographic information:

 Thomas A. Henzinger, Jalaj Upadhyay, and Shubham Upadhyay. Efficient private summation with applications to continual counting. In Proceedings of the 63rd IEEE Annual Symposium on Foundations of Computer Science (FOCS), 2023.